# Dietary Fibre from Whole Grains and Their Benefits on Metabolic Health

**DOI:** 10.3390/nu12103045

**Published:** 2020-10-05

**Authors:** Nirmala Prasadi V. P., Iris J. Joye

**Affiliations:** Department of Food Science, University of Guelph, Guelph, ON N1G 2W1, Canada; ijoye@uoguelph.ca

**Keywords:** dietary fibre, cereals, pseudo-cereals, chronic diseases

## Abstract

The consumption of whole grain products is often related to beneficial effects on consumer health. Dietary fibre is an important component present in whole grains and is believed to be (at least partially) responsible for these health benefits. The dietary fibre composition of whole grains is very distinct over different grains. Whole grains of cereals and pseudo-cereals are rich in both soluble and insoluble functional dietary fibre that can be largely classified as e.g., cellulose, arabinoxylan, β-glucan, xyloglucan and fructan. However, even though the health benefits associated with the consumption of dietary fibre are well known to scientists, producers and consumers, the consumption of dietary fibre and whole grains around the world is substantially lower than the recommended levels. This review will discuss the types of dietary fibre commonly found in cereals and pseudo-cereals, their nutritional significance and health benefits observed in animal and human studies.

## 1. Introduction

Consumers worldwide are interested in a healthy diet. Whole grain products, encompassing both cereals and pseudo-cereals, should constitute an important part of this healthy diet. The consumption of whole grain products is considered to have a beneficial effect on risk reduction of non-communicable diseases (NCD), including cardiovascular diseases, cancers, gastrointestinal disorders and type 2 diabetes [1,2,3]. It is widely accepted that it is through their high dietary fibre levels that these whole grain products play a very important role in the prevention and alleviation of NCDs. The term ‘whole grain products’ refers to products that are made with a relative proportion of bran, germ and endosperm tissue equal to what would naturally occur in intact grains [4]. 

According to the Cereals and Grains Association [5], ’whole grains consist of the intact, ground, cracked, flaked or otherwise processed kernel after the removal of inedible parts such as the hull and husk. All anatomical components, including the endosperm, germ, and bran must be present in the same relative proportions as in the intact kernel.’

Most cereal products that are currently on the market, however, are refined. Refined grain products are products that lack one or more parts of the integral kernel [6]. In the classical refining process for wheat e.g., the bran and the germ are separated from the starchy endosperm. The starchy endosperm is then further size-reduced to a fine white flour. Although during this process, the most functional part of a wheat kernel may be purified in the fine white flour to achieve the best quality end product, from a nutritional point of view, the refining process removes vital nutrients, dietary fibre and other phytochemicals from the other grain parts. As a result, the resulting refined products are of lower nutritional quality than the original whole grain products [7]. Therefore, the consumption of whole grain products, that have inherently a higher dietary fibre content than refined grain products and usually have a dietary fibre profile with a good balance between soluble and insoluble fibre components [8], undoubtedly can make a big difference in alleviation of risk on NCDs.

Guidelines on recommended daily intake of whole grain products vary among countries. Canada’s Food Guide [9] recommends replacing refined grain with whole grains, and the US dietary guidelines [10] recommend an optimal consumption level for wholegrain products of at least 85 g per day. The European Science Hub also emphasizes the importance of consumption of whole grain. Within the European Union, the different countries, however, have distinct guidelines [11]. The Swedish National Food Agency, e.g., recommends a daily consumption of about 70 g of whole grains for women, while 90 g is recommended for men. In Norway, a daily consumption of whole grain products is recommended, and the intake levels should reach 80 to 90 g/day [11]. According to research done by Micha and others [12], globally, the mean consumption of whole grain was only about 38 g/day. The values found on consumption of whole grain per country, however, varied widely, with values reported among 187 countries from 1990 to 2010, ranging from 1.3 to 334.3 g/day [12]. Overall, only 23 out of 187 countries displayed a mean whole grain consumption greater than 2.5 servings (~50 g) per day. The study indicated clearly that, on a global scale, whole grain consumption levels are far below the recommended levels (at least 2.5 servings/day). 

## 2. Dietary Fibre Present in Cereals and Pseudo-Cereals

### 2.1. Structure of Cereal Grains

Wheat, barley, oats and rye belong, as all true cereals do per definition, to the grass family (Poaceae). This family is a very diverse family, covering plants that humans have used to grow a lawn to plants that can grow several meters tall (bamboo) [13]. Cereal grains have a complex structure that is characterized by different cell layers. Although the individual structural parts of the different cereal kernels may differ significantly in terms of composition and size, the general cereal structure remains largely the same [8]. Three main parts can be distinguished: the embryo or germ, the endosperm, and the outer kernel layers that cover the embryo and endosperm or the so-called bran [14]. The starchy endosperm accounts for 80–85% of the grain. It is mainly composed of starch and protein. Bran and germ represent 12–18% and 2–3% of the dry grain weight, respectively [15]. The embryo is vital for the germination process, as it comprises the embryonic axis and scutellum. The embryo has the highest content of lipids and lipid-soluble vitamins of all fractions in the cereal kernel [16]. The endosperm has the highest economic importance. In the endosperm insoluble nutrients, mainly starch and proteins, are deposited as an energy source for the developing plant upon germination [15]. The aleurone layer is the outermost layer of the endosperm. It usually consists of 1 to 3 layers of cells, depending on the cereal. In some cases, pigmentation in the aleurone layer can give cereal kernels a distinct colour [16]. Although botanically, the aleurone layer is considered to be part of the endosperm, a major part of this layer has been shown to be removed during roller milling and, hence, is often not part of the refined white cereal flour [17]. The bran fraction consists of several different layers that can be distinguished from one another, i.e., outer pericarp, inner pericarp, testa, and nucellar epidermis (also called hyaline layer) (Figure 1) [15]. Inner and outer pericarp are rich in highly crosslinked polysaccharides, such as cellulose, lignin and heteroxylan [18,19]. The testa of barley, oats and rice is present as one cell layer, while wheat and rye testas generally have two distinct layers [14]. The nucellar epidermis is the maternal tissue that covers the endosperm, but is not prominent in all cereals [16]. In sorghum, this layer is very prominently present, but it is usually absent or only present as a thin layer in most other cereals [20]. 

### 2.2. Dietary Fibre Composition of Different Cereals

Dietary fibres are defined as “carbohydrates with a degree of polymerization of 3 or more that naturally occur in foods of plant origin and that are not digested and absorbed by the small intestine” [22]. Dietary fibre can be classified according to its water solubility in insoluble dietary fibre (IDF) and soluble dietary fibre (SDF) [23]. IDF includes cellulose, water-insoluble hemicellulose and lignin, and are mainly present in plants as structural cell wall components [24]. SDF consists of a variety of non-cellulosic polysaccharides and oligosaccharides. Examples are pectins, β-glucans and water-soluble gums [25]. SDF and IDF differ largely in their functionality as food ingredients and their physiological effects upon consumption [23]. With regard to the latter, SDF, by increasing the viscosity of stomach and intestinal contents, is believed to reduce the overall intestinal enzymatic activity, and to decrease post-prandial plasma glucose levels [26,27]. In addition, SDF are highly fermentable and increase the production of short chain fatty acids (SCFAs), which are important contributors in the management of CVDs [28]. IDF, conversely, mainly serves as a bulking agent and laxative, hence, increasing faecal mass and decreasing intestinal transit time [29]. A potential mechanism of IDF related to management of NCDs might be related to increased satiety and reduction in body weight [28]. Both SDF and IDF help prevent constipation, decrease re-adsorption of bile salts, and lower the risk of colon cancer [25].

Dietary fibre can be obtained from different dietary sources, which include grains, fruits and vegetables. The amount and composition of dietary fibre can vary with the source [23]. Cereals are an important source of dietary fibre, contributing to about 50% of the total dietary fibre intake in Western countries [30]. Vegetables deliver about 30 to 49% of the daily dietary fibre intake, while fruits contributed about 16%. Equal weights of fruits and vegetables contain less total dietary fibre relative to cereal grains, due to the higher moisture content. The proportion of IDF of the total dietary fibre varies depending on the type of fruit or vegetable that is studied [23]. Cellulose is the main component in the IDF fraction in plants, while pectin is a major fraction in the SDF fraction of fruits and vegetables [31].

#### 2.2.1. Dietary Fibre Composition of Wheat 

The total dietary fibre content of wheat ranges from 9 to about 20% (dry weight basis), and is composed of both insoluble and soluble fractions (Table 1) [32,33]. The cell walls of the starchy endosperm cells in wheat are composed of two major types of dietary fibre components; i.e., arabinoxylan (AX) and β-d-glucan. These cell walls may also contain small amounts of cellulose and glucomannans [13]. The cellulose content in wheat endosperm is usually very low (<5%) [32]. Cellulose is a linear polymer of β-(1-4) linked glucose units, that associates with other cellulose molecules to form a highly insoluble network [33]. 

Hemicellulose is a prominent type of DF in grains. Hemicellulose is defined as the non-cellulosic component in cell walls consisting of heterogenic polysaccharides [34]. Hemicellulose molecules can be grouped largely into four categories: xylans, xyloglucans, glucomannans and mixed linkage β-glucans [34]. Hemicelluloses can be soluble or insoluble, depending upon their size and structure (e.g., side chain substitutions and intermolecular crosslinks) [35].

AX and mixed linked β-glucan account for about 70% and 20% of the total dietary fibre content, respectively. AX molecules are composed of a linear backbone of d-xylopyranosyl (Xyl) residues linked through β-(1-4) glycosidic linkages (Figure 2). Residues of α-L-arabinofuranosyl (Ara) can be attached to the Xyl residues at the O-2 and O-3 positions (Figure 2). Four structural elements can, hence, be found in AX: non-substituted, O-2 or O-3 monosubstituted and disubstituted Xyl [36]. Ferulic acid can be esterified to arabinose residues on the O-5 position [37]. These ferulic acid structures can form bridges between AX chains, resulting in an increase of the AX molecular weight and a decrease in its water-extractability. 

A significant part of the AX (>30%) in wheat endosperm is present as a water-extractable (WE) fraction [59,60]. The water-unextractable AX (WU-AX) are typically crosslinked to other polysaccharides or lignin molecules in cell walls [61]. The structure of WU-AX is very similar to that of WE-AX, but the average molecular weight and (to some extent) the Ara/Xyl ratio are higher for WU-AX than for WE-AX [60] β-d-glucan has a relatively simple structure in cereals, as it is only built up of one monosaccharide, i.e., β-d-glucose, that can be linked through either β-1-4 or β-1-3 linkages [8]. 

Aleurone cells in wheat are characterized by thick cell walls. Relative to the starchy endosperm cell walls, the relative levels of AX and β-d-glucan in aleurone cell walls, however, remain the same [63]. AX in the aleurone layer, however, is highly esterified and crosslinked through diferulic acid bridges compared to starchy endosperm AX [64]. The pericarp cell wall composition is similar to the cell wall composition found in straw, characterized by highly branched AX. The AX in the pericarp also contain galactose and glucuronic acid residues and have a higher content of ferulic and diferulic acid residues [65].

#### 2.2.2. Dietary Fibre Composition of Barley and Oats

Barley is one of the earliest cultivated cereals and exists in hulled and hulless varieties. Hulled barley can be dehulled after harvest prior to processing [7]. However, the hull from barley is not that easy to remove, as it is ‘cemented’ to the outer layer of the kernel or caryopsis, i.e., the pericarp [43]. In both hulled and hulless types, the caryopsis is composed of the pericarp, testa (seed coat), aleurone layer, endosperm, and embryo [7]. Oats, on the other hand, is also a hulled cereal, but its hull is relatively easy to remove. Barley and oats are an excellent source of soluble and insoluble dietary fibre and other bioactive compounds. Soluble dietary fibre (mainly β-glucan) is located in the endosperm cell walls, while the (predominantly) insoluble dietary fibre fraction (cellulose, AX and lignin) is mainly found in the cereal bran [66]. The total dietary fibre content of dehulled barley and oats ranges from 10 to 28% [43,44] and 10 to 38% [23,41,42] (on dry matter basis), respectively (Table 1). Both barley and oats contain β-glucan as the primary non-starch polysaccharide in the whole kernel. AX is also found in both cereals, but in a much lesser content. β-glucan and AX are typically present as 70 to 20% of the total dietary fibre content in these cereals. Cereal β-glucan is composed of cellotriosyl and cellotetraosyl units linked through β-1-3 linkages (Figure 3) [7]. The presence of such β-1-3 linkages leads to bends in the polymer chain structure, allowing water to get in between the chains [7]. This explains the higher solubility of β-glucan as compared to cellulose, a structurally related polymer built exclusively of β-1-4-linked d-glucose units [67].

The β-glucan content in oats and barley varies with the genotype. β-glucan is distributed uniformly throughout the endosperm in barley, while it is more concentrated in the outer layers of oats endosperm [68]. Whole grain barley can provide similar amounts of β-glucan as oats do. Hulless barley varieties and barley varieties with low amylose content can even provide 1.5 to 4 times more β-glucan as compared to oats [7]. 

As endosperm cell walls of barley and oats are rich in β-glucan, the β-glucan content of barley and oats may not decrease with the removal of the outer bran layers [7]. The soluble β-glucan content even increases in the function of dehulling, indicating the dominant endosperm distribution of β-glucan [69].

#### 2.2.3. Dietary Fibre Composition of Rye

The dietary fibre content of rye is higher compared to wheat, as rye contains about 14 to 21% dietary fibre (Table 1) on dry matter base [42,70]. AX, cellulose, fructan and β-glucan are the dominant dietary fibre types in rye, with AX being the major dietary fibre component (i.e., 45% of total dietary fibre content) present in endosperm cell walls [70]. Although both rye and wheat contain AX, the content and solubility of AX in rye is higher compared to AX found in wheat [34].

Rye contains the highest amount of fructan among the here-discussed cereals. Fructan is a soluble dietary fibre composed of β-d-fructofuranosyl units, with or without terminal glucose residue [71]. Fructans of rye can be linear or branched in structure. A typical degree of polymerization of fructan in rye ranges anywhere from 2 to 60 [72]. 

The level of dietary fibre present in rye varies in function of its location within the kernel. The inner endosperm contains less dietary fibre (12%), while the outer endosperm and bran fraction contain about 22 and 38% dietary fibre on dry matter basis, respectively [73]. The higher levels of dietary fibre found in the outer kernel layers of rye are another illustration of the importance of eating whole grains.

#### 2.2.4. Dietary Fibre Composition of Other Grains

The dietary fibre content of rice (whole grain) varies from 2.7 to 9.9% (Table 1). This high variation in dietary fibre content is partially related to differences found in between rice varieties [34,47]. The dietary fibre content of brown rice is higher than the content found in white rice, in which, essentially, the outer kernel layers have been removed by abrasive milling. As is the case with the other cereals, the dietary fibre is also mainly found in the hull and bran of rice kernels [74]. In rice (whole grain), the major components of the IDF fraction are cellulose and water insoluble hemicellulose, while soluble AX and β-glucan make up the SDF fraction [75].

The dietary fibre content of corn varies between 3.7 and 19.9% on dry matter basis [42,76], of which IDF is the largest fraction (Table 1) [38]. Cellulose and hemicellulose are the main IDF fractions found in corn bran [76].

### 2.3. Structure of Pseudocereal Grains

Pseudocereals are largely underutilized crops that have recently been gaining attention due to the nutritional properties that are associated with them. Pseudocereals can be processed into, and used as, a flour, in a very similar way to the way in which wheat is processed and used. The three pseudocereals that have been most widely studied thus far are amaranth (*Amaranthus* spp; *Amaranthceae*), quinoa (*Chenopodium quinoa* subsp. *quinoa; Chenopodiaceae*), and buckwheat (*Fagopyrum esculentum; Polygonaceae*) [55]. These are dicotyledonous plants, as opposed to cereals like wheat, rice and barley, which are monocotyledonous [77]. 

Pseudocereal seeds are, similarly to cereals, also composed of several ‘layers’ (Figure 4). Perisperm, germ and endosperm are the three main areas containing food reserves in pseudocereals [76]. The kernel and ‘layer’ composition varies for the different pseudocereals. 

Amaranth is a small seed with diameters ranging from around 0.9 to 1.7 mm [76]. The major portion of the seed is the embryo, which is twisted in a circle. The embryo is large and encloses the perisperm and consists of the radicle and cotyledons, which is the main protein storage organ of the seed [78]. The seed coat is completely smooth and thin, and its colour can be white, cream, gold-yellow and even brown [55]. Quinoa kernels have the same structure as amaranth kernels. It produces small, spherical-shaped seeds, with diameters that vary between 1.0 and 2.6 mm. One gram of quinoa contains about 250 to 500 seeds [79]. Similar to what is the case for amaranth, the main storage tissues of quinoa seeds are the perisperm, embryo and cotyledons [80]. Buckwheat seed is pyramid-shaped with sizes ranging from 4 to 9 mm [79]. The seed is covered with a dull brown or grey pericarp that is tightly attached to the seed [81]. The embryo is embedded in the centre of the endosperm and has two cotyledons [82]. 

### 2.4. Dietary Fibre Composition of Pseudocereals

Amaranth, quinoa and buckwheat are pseudocereals with a long history of utilization as food ingredients and have very interesting nutritional characteristics. Pseudocereals have been gaining popularity in the last decade as ingredient for gluten-free products. Their use substantially increases the dietary fibre content of these products, which are typically deficient in dietary fibre [49]. Although the dietary fibre content varies between amaranth species, the total dietary fibre content of amaranth varies between 9 and 21% (dry weight basis) [83,84], while quinoa contains around 7 to 21% TDF [50,52,53]. Based on a monosaccharide analysis of dietary fibre extracted from amaranth and quinoa samples, the dietary fibre in these pseudocereals is mainly composed of galacturonic acid, arabinose, xylose, glucose and galactose. According to this monosaccharide composition and analysis of the linkages, the dominant fraction of both the soluble and insoluble dietary fibre in these pseudocereals is classified as pectic polysaccharide [85]. Xyloglucans are the second most prominent dietary fibre present in amaranth and quinoa whole grains. Cell walls of amaranth also contain significant amounts of phenolic acids. High levels of ferulic acid were found, while coumaric acid and caffeic acid are also present, but in lower amounts [78]. Both quinoa and amaranth have a high proportion (~22% of the total dietary fibre content) of SDF compared to wheat (about 15%), indicating promising potential with regard to colon health [86]. 

The total dietary fibre content of buckwheat groats (7–11.9%) is lower than the dietary fibre content found in most cereals such as wheat, barley and oats (Table 1). The majority (~70%) of dietary fibre from buckwheat groats is water insoluble [82]. The water soluble fibre of buckwheat seeds is mostly classified as pectin, arabinogalactan and xyloglucan [81]. Pectin was found in the cell walls of the outer and inner epidermis, and the endosperm of buckwheat seeds [81].

## 3. Dietary Fibre from Whole Grains and Health Implications 

Many health benefits associated with the consumption of whole grains are ascribed to the higher dietary fibre content in whole grain ingredients, as opposed to the levels found in refined cereal ingredients. Whole grains provide a good balance of soluble and insoluble fibre (Table 1). 

A meta-analysis done by Huang et al. [87] showed that a high consumption of whole grains or cereal fibre could be related to a reduced risk of NCDs.

### 3.1. Cardiovascular Health

Atherosclerosis and subsequent cardiovascular complications, such as myocardial infarction, stroke, and heart failure, are leading causes of death worldwide [88]. Multiple factors have been identified as risk factors for the development of CVDs. These risk factors include both non-modifiable risk factors (e.g., age, gender, and family history) and modifiable risk factors (e.g., lipid profile, blood pressure, hyperglycemia etc.) [89]. According to Health Canada, heart diseases are the second leading cause for deaths in Canada [90]. The beneficial effects of dietary fibre consumption on CVD protection have been well-documented in both animal and human studies. The protective effect of dietary fibre on CVDs may be due to the ability of SDF to form viscous gels. Increasing the viscosity of the gut content reduces the reabsorption of bile acids which, on its turn, reduces the circulation of cholesterol in blood. SDF also triggers the formation of short chain fatty acids (SCFA) by colon bacteria [91]. These SCFAs include butyrate, acetate and propionate and have different functional properties in the human body. Butyrate, for example, is known to reduce the progression of atherosclerosis, while propionate is known to inhibit cholesterol synthesis and its accumulation in the liver [92]. High blood cholesterol levels were identified as a risk factor for the incidence of CVDs [62]. Cereal β-glucan is one of the main types of SDF that have an effect on CVD prevalence. After considering the scientific evidence, the Food and Drug Administration (FDA) allowed a health claim that soluble fibre from oats may reduce the risk of heart diseases [93].

Several studies using animal models have indicated that AX have the potential to reduce blood serum triglyceride and cholesterol levels, both considered to be important modifiable risk factors for CVD [94]. One such study focused on hypercholesterolemic hamsters. Supplementation of the feed with extracted AX from wheat bran (alkaline extraction) reduced the total and LDL cholesterol levels in these hamsters [95]. Moreover, the AX supplementation led to increased propionate and total short-chain fatty acids (SCFAs) concentrations in the colon [66].

In another study, five week old mice were fed with a lard based high fat diet for one week, after which the diet was substituted with enzyme treated wheat bran (wheat bran was treated with xylanase and cellulase to increase the soluble AX content). A significant reduction in body weight and liver triglyceride content was observed with the administering of enzyme treated wheat bran [96]. This supplementation has also altered the gut microbiota composition [7]. In a rat study, a decrease in plasma cholesterol levels and an increase in SCFA levels in portal vein was found when refined wheat flour was substituted with whole wheat flour in the diets that were fed to the rats [97]. Research on pseudocereals carried out by Konishi et al. [98] reported that a diet supplemented with quinoa pericarp significantly reduced liver cholesterol in mice. Similarly, a cholesterol lowering effect was also found when hypercholesterolemic rabbits were fed with extruded amaranth products [99]. In these rabbits, reductions in the levels of total cholesterol, LDL-cholesterol, VLDL-cholesterol, and triacylglycerols in plasma were observed after feeding a diet with extruded amaranth for 21 days [67]. 

According to a study carried out with 34,492 postmenopausal women that were monitored for 6 years, a higher intake of whole grains was associated with a reduced death rate due to CVD. In another experiment using a nurses’ health study, a strong inverse relationship between whole grain consumption and risk of CVD was observed [100]. Another cohort study evaluated the association between the dietary fibre type and cardiovascular risk factors in a Spanish working population [101]. This study pointed to an inverse relationship between IDF intake and total cholesterol and blood pressure, while SDF intake inversely affected the triglyceride content [101] when analysing the blood samples. 

In an intervention study, a diet rich in barley β-glucan was administered to mildly hypercholesteremic subjects (25 subjects) [102]. A reduction in total plasma cholesterol content was observed for the test population that was fed a diet containing 3 to 6 g of β-glucan from barley compared to those subjects that were administered a diet that did not contain β-glucan [102]. In another study, β-glucan was administrated in three levels, i.e., low (only whole wheat flour), medium (50:50 whole wheat flour: barley flour) and high (only barley flour) β-glucan level. Total cholesterol levels were 4, 9, and 10% down, respectively, after consuming the low, medium and high β-glucan diets compared to what was detected for the subjects that were fed a control diet [103]. A meta-analysis performed on 28 randomized control trials that used at least 3 g of oat β-glucan per day to evaluate cholesterol lowering effects concluded that these levels of oat β-glucan can reduce both low density lipoprotein (LDL) and total cholesterol levels. The above specified β-glucan dose, on the other hand, did not shown any significant impact on the level of high-density lipoproteins (HDL) [104]. 

A high dietary fibre intake has also been shown to affect the incidence of hypertension, another risk factor associated with CVDs [105]. Several studies suggested that the consumption of SDF provides a safe way to reduce blood pressure [106,107,108]. In a Mediterranean cohort study, an inverse relation was found between cereal fibre intake and the risk of hypertension [109]. Another cohort study was evaluating the effect of the type of dietary fibre on risk factors in a Spanish working population and found an inverse relationship between IDF consumption and systolic and diastolic blood pressure [101]. 

In a randomized parallel group study involving hypertensive men and women, eating either a low fibre cereal diet or a high fibre oat meal, a significant reduction in systolic and diastolic blood pressure was observed for the high fibre diet consuming test subjects [110]. In a study to understand the effect of dietary fibre type on blood pressure reduction, a significant reduction in blood pressure was observed for food containing higher total dietary fibre and IDF levels [105]. However, additional studies are still needed to confirm these findings [111].

### 3.2. Blood Glucose Levels

Diabetes is a chronic condition during which the human body loses its ability to produce or properly use insulin. There are three common types of diabetes: type 1, type 2 and gestational diabetes. About 90% of the diabetes cases in Canadian adults are identified as type 2. Type 2 diabetes is a metabolic disorder caused by a low insulin production by the pancreas [112]. Among many other factors, the diet has been identified as an important aspect in managing type 2 diabetes. Diabetes or elevated glucose levels in blood are also considered risk factors for CVDs [89].

Dietary fibre from cereals was found to be effective in reducing the postprandial blood glucose response (i.e., blood glucose levels after a meal) and increasing the insulin response [113]. As outlined above, soluble dietary fibre has the capacity to increase the viscosity of stomach content. An increase in the viscosity of the gut content caused by cereal dietary fibre plays a major role in reducing glucose absorption [27]. The higher viscosity also slows the gastric emptying process down and reduces the rate of starch digestion (and associated mono- and disaccharide production) and, hence, causes a delay in glucose absorption [114]. The slower digestion rate can also be the result of the formation of a ‘thick layer’ around the food bolus, that reduces the access digestive enzymes have to the inner part of the food bolus and hinders contact with absorbing surfaces in the gastrointestinal tract [115,116]. Even though SDF are considered to display anti-diabetic effects, it has been found that IDF are more potent in reducing the risk of type 2 diabetes [117,118]. It is believed that fermentation and secondary metabolites play a role in the mentioned effect [119]. The mechanisms behind these benefits of IDF need to be studied further. 

Studies on pseudocereals, such as amaranth, showed that its consumption may also reduce the blood serum glucose levels and increase insulin levels in diabetes-induced rats [120]. When fed extruded amaranth snacks with a high fibre content, a reduced glycemic response was observed, relative to what was measured for a control population that was fed white bread [78]. However, some studies have shown that amaranth consumption could actually increase the glycemic response [78,121]. This study compared the in vitro starch digestibility of processed amaranth seeds to that of white bread. Cooked, extruded, and popped amaranth seeds had a starch digestibility similar to that of white bread while flaked and roasted seeds generated a slightly increased glycemic response [80]. These opposing results may be related to differences in the processing conditions and the associated changes induced in the dietary fibre fraction and starch in the different studies. 

According to a meta-analysis of a cohort study done by Schulze and others, the intake of dietary fibre of cereal origin associates inversely with the risk of diabetes development [118]. Another study focusing on the association between intake of dietary fibre and type 2 diabetes is the European Prospective Investigation into Cancer and Nutrition (EPIC)-InterAct study [122]. This study revealed that there is an inverse relationship between the total and cereal dietary fibre consumption and the risk of developing type 2 diabetes [122].

Cereal β-glucan has been studied extensively to evaluate its effect on blood glucose responses after consuming a meal. In a study carried out with test subjects with and without type 2 diabetes, a meal with native cell wall fibre from oat bran and extracted oat gum reduced the postprandial blood glucose levels compared to a wheat farina meal [123]. In another study, a meta-analysis performed using four articles studying a total of 350 subjects, a relation was found between the consumption of oat β-glucan and a reduction of plasma glucose levels [124]. Consumption of AX has also been related to a significant reduction of the postprandial glucose response [114]. Soluble AX was found to reduce the rate of the gastric emptying process, hence reducing the absorption of glucose [125]. Plasma glucose levels were measured after administering AX-enriched and control white bread, in a double-blind crossover study design using human subjects. Postprandial glucose concentrations observed after AX-enriched meals were significantly lower than what was found for those subjects that were eating the control meal [126]. Similar to what was outlined above for β-glucan, the viscosity increasing effect of AX exerted beneficial effects on glycemia and insulinemia (i.e., presence of high concentration of insulin in blood) [127]. As AX is a major component of the dietary fibre fraction in many cereals, the consumption of any type of whole cereal grain is believed to unlock the health benefits that are associated with AX [128]. 

The effect of IDF on prevention of diabetes has also been recognized in recent studies. Even though the mechanism is not yet fully understood, the effect may be related to an increased satiety and changes in body weight [129]. The consumption of IDF levels according to the recommended levels has helped to accelerate the early insulin response, and is also associated with a significant reduction in postprandial glucose value [28]. Large cohort studies pointed to the possibility that IDF from cereals may reduce the risk of developing type 2 diabetes [129,130]. Researchers have been observing, however, largely contrasting results in this field, clearly illustrating the need for more studies [131].

### 3.3. Gastrointestinal Health

Dietary fibre can positively affect gastrointestinal health. Whole grains are rich in dietary fibre and usually have a lower energy density. Dietary fibre also plays a vital role in providing a suitable environment for gut microbiota by acting as prebiotics. Prebiotics are ingredients that are resistant to gastric acidity and hydrolysis by enzymes. They can be fermented in the colon, hence, changing the composition of the gut microbiota [132]. 

According to the Canadian Cancer Society, colorectal cancer is recognized as the third most commonly diagnosed cancer in Canada [133]. Many studies thus far have focused on finding a correlation between the prevalence of colorectal cancer and the consumption of dietary fibre. A strong theoretical base to prove this correlation exists as dietary fibre dilutes faecal carcinogen and procarcinogen concentrations. In addition, it also reduces the residence time of the carcinogens in the lower gastrointestinal tract, reducing their absorbance [90]. Dietary fibre present in whole grains can be degraded by bacterial enzymes in the human colon to produce SCFA. SCFA have shown to exhibit protective effects against the growth of tumour cells [134]. According to a meta-analysis done to evaluate the association between dietary fibre and colorectal cancer incidence, an increase of the cereal dietary fibre intake with about 10 g/day is associated with a 9% decrease in risk on colorectal cancer [135].

Inflammatory bowel disease (IBD) is a chronic disease characterized by a painful inflammation of the small and large intestine. The most commonly occurring IBDs are Crohn’s disease and ulcerative colitis [136]. Crohn’s disease can cause inflammation in any part of the gastrointestinal tract, while ulcerative colitis causes inflammation in the large intestine [137]. Studies have described the positive effects of dietary fibre present in whole cereal grains on gut microflora. One study, for example, positively evaluated the effect of oats β-glucan on Crohn’s disease in rats [138]. The presence of various types and lengths of dietary fibre was found to be crucial to these positive effects [136]. According to macroscopic and microscopic analysis, both low and high molecular weight β-glucan are able to reduce the macroscopic and microscopic lesions occurring in mucosal and submucosal layers of the colon in rats with induced colitis [136]. β-glucan supplementation of the diet increases the levels of beneficial lactic acid bacteria and increases the SCFA content in rat faeces. This will reduce the growth of pathogenic microflora and the production of toxic metabolites in the colon [136]. Although both high and low molecular weight β-glucan were shown to exert positive effects, the effectiveness in inhibiting mucosal infections was higher for the high than the low molecular weight β-glucan samples [136]. Dietary fibre has shown to attenuate experimental colitis in animal models. However, some large cohort studies could not support the claimed involvement of dietary fibre in the prevention of ulcerative colitis [139,140]. This clearly indicates the need for future work to investigate long term effects of the consumption of dietary fibre types on IBD. 

The consumption of dietary fibre plays a very important role in maintaining healthy gut microbiota [141]. The symbiotic relationship among gut microbiota and the human being plays a crucial role in risk reduction of NCDs, including IBD and colorectal cancers [142]. Dietary fibre helps in preserving the diversity of gut microbiota [143]. A lack of dietary fibre in our daily diet reduces the diversity of this gut microbial community [144]. Similarly, due to this diverse nature, the human gut microbiome contains diverse microbial genomes which produce thousands of enzymes targeting dietary fibre [145]. This results in the formation of a number of metabolites from dietary fibre conversion in the GIT, including SCFAs that are believed to help managing NCDs [144].

Another well-known function of dietary fibre is promoting laxation and preventing constipation. An increased intake of dietary fibre can indeed prevent and/or manage the prevalence and severity of constipation and haemorrhoids [146]. Wheat bran and high fibre cereal fractions are commonly recommended to prevent constipation. The water holding capacity of these dietary fibre fractions plays an important role in the laxative properties, as it leads to faecal bulking. According to Cummings [147], the ability of dietary fibre to result in faecal bulking varies with the dietary fibre type. As an example, the faecal weight increase per gram of administered wheat bran is about 5.4 g, while whole oats increase the faecal weight by 3.4 g per gram of administered oats in human studies. 

### 3.4. Obesity and Weight Management

There is strong epidemiological evidence linking the consumption of dietary fibre (more specifically the intake of whole grains) to overweight and obesity mitigation. The main effect behind this effect, may be a reduced appetite and prolonged satiety feeling after consumption of these products [148]. An inverse association was observed between wholegrain consumption levels, on the one hand, and body mass index (BMI) and risk of overweight and obesity in men as well as women, on the other hand, in a cohort study carried out in the Netherlands [149]. The association in men was found to be stronger than the association found for the female participants [149]. A cross-cultural study of 16 cohorts in seven countries showed that the BMI and subscapular skin fold thickness were inversely associated with total dietary fibre intake, suggesting that a reduced intake of fibre was an important determinant in the storage of body fat [150]. 

Studies have demonstrated that β-glucan consumption can enhance the postprandial satiety feeling, and reduce the body weight, BMI and total energy intake [151,152]. In another study, a meta-analysis was done using 20 studies to evaluate the effect of cereal β-glucan consumption on body weight, BMI and anthropometrics [153]. The study confirmed that the consumption of β-glucan from cereal sources leads to a significant reduction of body weight and BMI. Possible mechanisms for the prolonged satiety feelings may be linked to the gel forming ability of soluble β-glucan and other soluble fibres, and the bulking effect of insoluble fibres [154]. In addition, a release of appetite suppressants such as cholecystokinin was shown in response to the consumption of β-glucan at a minimum dose of 3.8 g per day, in a study done using 14 human subjects (7 male and 7 female) [155].

### 3.5. Undesirable Effects Associated with Consumption of Dietary Fibre

Although the above suggests that dietary fibre is exclusively associated with beneficial effects on human health, the intake of dietary fibre may also cause negative effects on mineral and overall micronutrient absorption [24]. Several in vitro studies have shown that both IDF and SDF display mineral binding properties to various extents [156]. Mineral binding properties vary with the type of fibre, concentration and with pH and ionic strength [157]. Similarly, dietary fibre may alter the bioavailability of vitamin [158]. The bioavailability of different kinds of vitamin B was studied, and it was observed that their bioavailability can vary with the characteristics of dietary fibre (type, molecular weight and the content [159]. Research suggests that various types of dietary fibre (hemicellulose, lignin and pectin) reduce the bioavailability of β-carotene in human subjects [160]. 

## 4. Conclusions 

Persons who consume a higher number of servings of whole grain foods as a source of dietary fibre are at lower risk for developing coronary heart diseases, diabetes, obesity and certain gastrointestinal disorders. However, even though the benefits of dietary fibre consumption are documented well, the consumption of dietary fibre is still below the recommended levels. The same is true for the consumption of whole grain products. 

Whole grains of cereals and pseudocereals contain a wide variety of dietary fibre types. Some examples of cereal dietary fibre types are arabinoxylan, β-glucan, xyloglucan, pectic polysaccharides and fructan. Cereal dietary fibre exists as both soluble and insoluble dietary fibre fractions. Different cereals have typically a different dietary fibre profile. Wheat grain e.g., is rich in AX, while barley and oats are recognized for the functional properties associated with their most important dietary fibre type, i.e., β-glucan. One of the main effects of soluble dietary fibre is increasing the viscosity of the gut content. In contrast, insoluble dietary fibre absorbs more water and helps in faecal bulking. 

Benefits of dietary fibre from cereals and pseudo-cereals have been studied throughout the years. Intake of some of the dietary fibre types such as oat β-glucan have been recommended due to approved health benefits, while many other types of dietary fibre are still studied for their specific effects. Because these health benefits are interconnected, often synergistic and individu-specific, it is difficult to obtain solid evidence of the health effects per dietary fibre. However, more research and communication on these health benefits is needed to translate the science behind these beneficial effects into useful information for broader public health advice for people seeking healthy eating patterns. 

## Figures and Tables

**Figure 1 nutrients-12-03045-f001:**
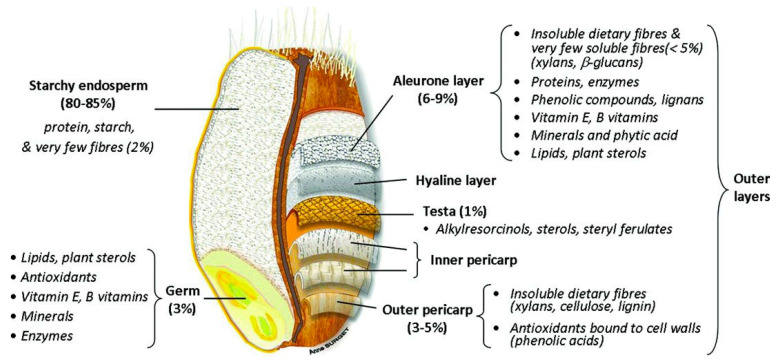
Wheat grain structure [21].

**Figure 2 nutrients-12-03045-f002:**
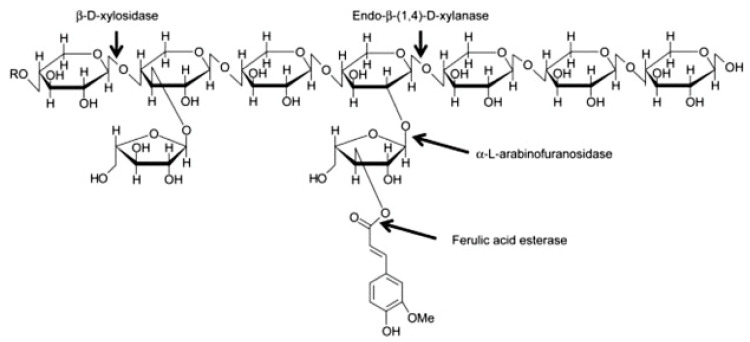
Structure of arabinoxylan [62].

**Figure 3 nutrients-12-03045-f003:**
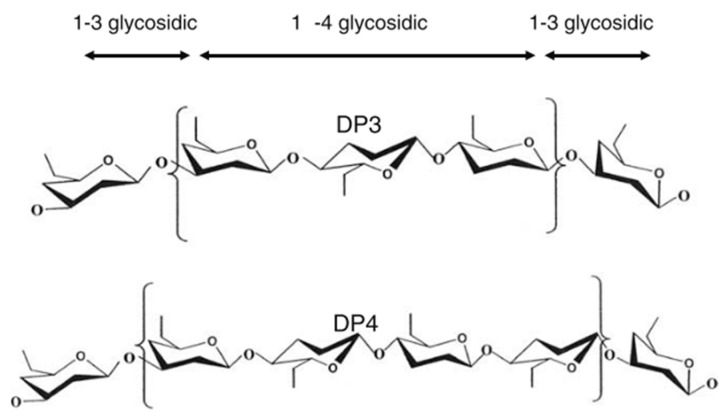
Molecular structure of cereal β-glucan [68].

**Figure 4 nutrients-12-03045-f004:**
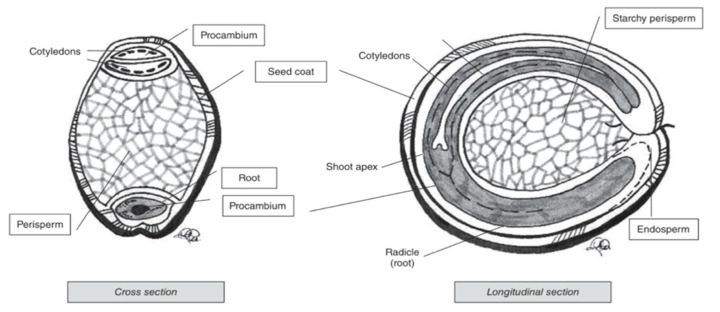
Cross and longitudinal section of amaranth seed [76].

**Table 1 nutrients-12-03045-t001:** Dietary fibre content (total, insoluble and soluble) of cereals and pseudo-cereals (g/100 g).

	TDF	IDF	SDF	Reference
Wheat (*Triticum aestivum* L., *Triticum durum Desf*.)	11.6–17.0	10.2–14.7	1.4–2.3	[38]
	10.2–15.7	7.2–11.4	1.9–2.9	[39]
	9.2	-	-	[40]
Oat (*Avena sativa* L.)	13.7–30.1	-	11.5–20.0	[41]
	10.3	6.5	3.8	[23]
	11.5–37.7	8.6–33.9	2.9–3.8	[42]
Barley (*Hordeum vulgare* L.)	14.6–27.1	12.0–22.1	2.6–5.0	[42]
	16.8–27.9	-	-	[43]
	10.1	-	-	[44]
Rye (*Secalecereale* L.)	15.2–20.9	11.1–15.9	3.7–4.5	[32]
	14.7–20.9	10.8–15.9	3.4–6.6	[45]
Rice (*Oryza sativa* L.)	9.9	5.4	4.4	[46]
	2.7–4.9	1.9–4.2	0.6–1.1	[47]
Corn (*Zea mays* L.)	3.7–8.6	3.1–6.1	0.5–2.5	[48]
	13.1–19.6	11.6–16.0	1.5–3.6	[42]
Amaranth (*Amaranthus* spp.)	8.9–20.6	-	-	[49]
	11.4	7.7	3.7	[50]
	11.8	9.1	2.7	[51]
Quinoa (*Chenopodium quinoa Willd.*)	7–9.5	4.9–5.6	2.1–3.9	[50]
	16.2–21.6	-	-	[52]
	11.6–15.1	9.9–12.2	0.4–2.9	[53]
Buckwheat (*Fagopyrumesculentum Moench.*)	7.0	2.2	4.8	[54]
	11.9	5.8	6.1	[55]
Teff [*Eragrostis tef* (*Zucc.*) *Trotter*]	4.54	-	0.85	[56]
Sorghum (*Sorghum bicolor*)	7.55–12.3	6.52–7.90	1.05–1.23	[57]
Millets (*Eleusine coracana* (L.) *Gaertn.*)	13.0–13.8	12.5–13.2	0.52–0.59	[58]

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
