# Peer review of "Dietary Fibre from Whole Grains and Their Benefits on Metabolic Health"

_nutrients, 2020, doi:10.3390/nu12103045_

Round 1

Reviewer 1 Report

This is a useful review, which provides a good (qualitative) overview of the different dietary fibre fractions associated with cereals and pseudo-cereals.  This is useful, as there is a tendency (particularly in some of the nutrition literature) to over-simplify this (and suggest dietary fibres/whole grains are more homogenous in their composition).

There is some useful information about how some of these dietary fibre fractions may play key roles in reducing several important non-communicable diseases.

There are a few suggestions (below) for how this review could potentially be a little stronger:

Is there any merit in a brief mention of how dietary fibres in whole grains may be similar or differ from dietary fibres found in other plant material (e.g. in fruits and vegetables)?

Although there is a brief mention of 'prebiotics' and 'gut microflora', and considering the current high levels of interest (and publications) linked to the 'gut microbiome'; is there an opportunity to have some more detailed discussion about how wholegrain dietary fibres may be influencing NCDs via the gut microbiome?

The review's focus is on the health benefits of whole grain dietary fibres, but is it worth mentioning any possible negatives associated with whole grain consumption (e.g. any issues about mineral/micro-nutrient binding asnd bioavailability)?

In some parts of the world there is some public interest in low carbohydrate diets, which may have implications for reducing whole grain consumption - is this worth a brief mention?

Reviewer 2 Report

The authors present a review of the health benefits of cereal and pseudo-cereal fiber.

The abstract appears to be a bit incoherent, as the focus is clearly on cereal fibers, but as typical examples are given arabinoglucans, fructanes and pectins, but not cellulose, hemicellulose or ß-glucan. Surprisingly, neither the uncommon ones, nor the very important cellulose or hemicellulose are part of the review.

Also, the title suggests are broader view on "health" rather than just metabolic health. It is recommended to clarify your focus; what about inflammation and inflammatory diseases (autoimmune diseases? asthma? allergies?), mental health, etc.?

The introductory section, showing the relative abundance of different types of fiber in different types of grains, is very well written. Comparison of fiber contents is a key message for the understanding; thanks for showing this data.

In order to boost the clarity of this section, I recommend adding a table, showing all percentages/ranges for soluble / insoluble fiber, specific types of fiber etc. for each type of grain. This table should also include some less common types of grain, which are currently "hyped" as super-foods (millet (sorghum, teff...), spelt...) in order to show, that their fiber profile is not so different compared to more common types of grain and the hype is not justified.

Similarly, the review (on fiber content and health effects) does not at all address rice and maize, which are very common types of grain.

When addressing different health outcomes, a consistent hierarchy of evidence should be followed step-by-step: pre-clinical evidence, cohort studies, intervention studies. Lack of evidence should be pointed out without hesitation. There is clearly a huge need for more research in this field.

Also, cardiovascular health is not identical with low cholesterol levels. Effects on blood lipids and other risk factors should be presented separately from "hard outcomes" such as MI, stroke, mortality etc.

Also, there is no mentioning of effects on blood pressure, which is also a major CVD risk factor.

Furthermore, a clearer distinction between soluble and insoluble fibers is necessary, as strong epidemiological evidence for long-term risks (T2DM, CVD, cancer) is shown predominantly for insoluble fiber. In this context, differences in fermentability should also be addressed more thoroughly. Insoluble fiber is mostly poorly fermentable, requiring a different explanation for metabolic benefits other than the production of SCFA.

For both CVD and glucometabolic health, there is no long-term data supporting soluble fibers (including ß-glucans; Reynolds et al. 2020). On the other hand, some pioneer trials on insoluble fiber (cellulose, hemicellulose) have investigated effects on insulin resistance and diabetes risk, indicating a potential benefit (Weickert et al. 2011, Hattersley et al. 2013, Honsek et al. 2018, Kabisch et al. 2019) mirroring the evidence from cohort studies.

Round 2

Reviewer 2 Report

Thanks a lot for your revision, which significantly improved the manuscript. As there a still left some points of criticism, I kindly ask you for a deeper look in the following aspects:

The term "hemicellulose" should at least be introduced as overarching category for AX, AG, XG and other types of fiber. Pectins and fructans are negligible fractions in whole grain and should not be part of either abstract or manuscript. 

Once again: When addressing different health outcomes, a consistent hierarchy of evidence should be followed step-by-step in the presentation: pre-clinical evidence (if only; if necessary), cohort studies, intervention studies. In some chapters, the logic thread still gets lost. 

As an example, for diabetes prevention, no cohort studies are mentioned at all (Schulze et al. 2007, InterAct et al. 2015). However, these studies point out, that predominantly insoluble cereal fiber bears a potential for risk reduction, while soluble fiber does not. From this point of view, the first large interventional studies on IDF in humans (e.g. Weickert et al., Honsek et al., Kabisch et al.) need to presented in a more extended manner. For SDF, SRMAs such as the Reynolds paper and earlier works are an excellent way to avoid cherry-picking single articles with strong effects.

Lack of evidence should be pointed out without hesitation. There is clearly a huge need for more research in this field. You cited Reynolds et al. [114] as prove for antihyperglycemic effects, however, this SRMA concludes, that especially in long-term perspective the evidence for soluble fiber is not there. 

Once again, I suggest to clearly discriminate between cardiovascular health ("hard outcomes" such as MI, stroke, mortality etc.) and mere surrogates and risk factors (cholesterol, blood pressure). There is only epidemiological evidence for CVD protection. Interventional studies on soluble fiber show an effect on LDL, but this does not automatically translate into reduced CVD risk. Same is true for BP, inflammation and antidiabetic effects.

Thus, the clearer distinction between soluble and insoluble fibers is not only necessary in the introduction, but also when summarising epidemiological and clinical evidence. There is strong cohort evidence and first clinical evidence for diabetes prevention for IDF, but not SDF. There is large mechanistic data supporting antidiabetic effects of SDF (incretins, microbiome, SCFA), but not for IDF. This contradiction needs to be highlighted.
